materials science/nanotechnology/biomaterials

carboxymethyl chitosan, MCM-41, clindamycin, composite hydrogels, antibacterial activity, osteogenic activity

**Authors for correspondence:**
Weerachai Singhatanadgit
e-mail: s-wrch@staff.tu.ac.th
Wanida Janvikul
e-mail: wanidaj@mtec.or.th

This article has been edited by the Royal Society of Chemistry, including the commissioning, peer review process and editorial aspects up to the point of acceptance.

# Antibacterial and osteogenic activities of clindamycin-releasing mesoporous silica/carboxymethyl chitosan composite hydrogels

Piyarat Sungkhaphan[1], Boonlom Thavornyutikarn[1], Pakkanun Kaewkong[1], Veerachai Pongkittiphan[1], Soraya Pornsuwan[2], Weerachai Singhatanadgit[3] and Wanida Janvikul[1]

[1]Biofunctional Materials and Devices Research Group, National Metal and Materials Technology Center, Pathumthani, Thailand
[2]Department of Chemistry and Center of Excellence for Innovation in Chemistry (PERCH-CIC), Faculty of Science, Mahidol University, Bangkok, Thailand
[3]Faculty of Dentistry and Research Unit in Mineralized Tissue Reconstruction, Thammasat University (Rangsit Campus), Pathumthani, Thailand

PS, 0000-0003-1949-7531; BT, 0000-0002-9814-0065;
PK, 0000-0002-3306-1390; SP, 0000-0003-4756-7891;
WS, 0000-0001-7594-1305; WJ, 0000-0002-2560-0075

Conventional treatment of jaw bone infection is often ineffective at controlling bacterial infection and enhancing bone regeneration. Biodegradable composite hydrogels comprised of carboxymethyl chitosan (CMCS) and clindamycin (CDM)-loaded mesoporous silica nanoparticles (MCM-41), possessing dual antibacterial activity and osteogenic potency, were developed in the present study. CDM was successfully loaded into both untreated and plasma-treated MCM-41 nanoparticles, denoted as (p)-MCM-41, followed by the incorporation of each of CDM-loaded (p)-MCM-41 into CMCS. The resulting CDM-loaded composite hydrogels, (p)-MCM-41-CDM-CMCS, demonstrated slow degradation rates (about 70% remaining weight after 14-day immersion), while the CDM-free composite hydrogel entirely disintegrated after 4-day immersion. The plasma treatment was found to improve drug loading capacity and slow down initial drug burst effect. The prolonged releases of CDM from both (p)-MCM-41-CDM-CMCS retained their antibacterial effect against *Streptococcus sanguinis* for at least 14 days *in vitro*. *In vitro* assessment of osteogenic activity showed that the CDM-incorporated composite hydrogel was

cytocompatible to human mesenchymal stem cells (hMSCs) and induced hMSC mineralization via p38-dependent upregulated alkaline phosphatase activity. In conclusion, novel (p)-MCM-41-CDM-CMCS hydrogels with combined controlled release of CDM and osteogenic potency were successfully developed for the first time, suggesting their potential clinical benefit for treatment of intraoral bone infection.

## 1. Introduction

Jaw bone infection, such as osteomyelitis, infected osteoradionecrosis (ORN) and medication-related osteonecrosis of the jaw (MRONJ), has been a major challenge in public healthcare with serious social and economic implications. Treatment of jaw bone infection is complicated by persistent exposure to oral bacteria and insufficient vascularization. The most commonly reported microorganisms are obligate or facultative anaerobic species, such as viridans streptococci and staphylococci [1]. Surgical treatment with prolonged (weeks to months) antibiotic therapy is normally required [1]. Bacterial infection is also noted to have some role in the development of MRONJ [2]. Moreover, bone infection is often associated with abnormal bone remodelling due to a defective bone repair process at the infection site. Conventional treatment is frequently ineffective at controlling the infection and enhancing bone regeneration, with an alternative approach being required.

Local delivery of a biomaterial with dual antibacterial activity and osteogenic potency is expected to simultaneously treat bone infection with minimized side effects of systemic use and enhance bone regeneration. Among the recommended antibiotics for jaw bone infection, clindamycin (CDM) has several advantages [3]. Two most commonly used antibacterial agents, penicillin and metronidazole, showed lower bone penetration profiles than CDM [4], which is highly active against gram-positive and obligate/facultative anaerobic bacteria [5]. CDM has had a long-standing role as a first line agent in penicillin-allergic patients to treat infections and abscesses. Unlike penicillin, CDM very rarely causes allergy [6]. However, a major disadvantage of CDM is its tendency to cause antibiotic-associated diarrhea following oral route administration [7]. A relatively high dose and duration of oral administration of CDM, i.e. 1200–1800 mg day$^{-1}$ for longer than 7 days [8], increases the risk of development of antibiotic resistance [9]. Moreover, bone loss associated with antibiotic-induced intestinal microbial imbalance (dysbiosis) has previously been reported in patients following oral antibiotic therapy [10,11]. An appropriate use of antibiotics in bone infection, therefore, helps lower the risk of the development of microbial drug resistance [12]. Recently, a local application of antimicrobial agents is recommended for the prevention and treatment of oral tissue infection in MRONJ [2]. This emphasizes the importance of effective local delivery of antibacterial agents for bony infection. The development of efficient biocompatible antibacterial-releasing carriers is, therefore, undoubtedly beneficial.

Hydrogels for drug delivery systems play an important role in the delivery of drugs or active agents to the therapeutic target/disease locations [13,14]. Generally, hydrogel materials for drug delivery systems need to be biodegradable, biocompatible and non-toxic. Sodium carboxymethyl chitosan (CMCS), a water-soluble derivative of chitosan, has been widely employed as a hydrogel material in various biomedical applications [15,16], owing to its desirable properties such as biodegradability, biocompatibility, antimicrobial activity and ease of gel-formation [17]. Attempts to fabricate CMCS-based hydrogels for therapeutic drug delivery have been reported. For example, Agarwal et al. [18] fabricated gelatin and CMCS-based scaffolds and used them as therapeutic delivery vehicles for wound healing treatments. Additionally, Gao et al. [19] developed implantable hybrid hydrogels from CMCS and doxorubicin (DOX)-loaded gelatin nanoparticles for the local treatment of solid tumours. However, there still exist major drawbacks to using polymeric hydrogels, e.g. low mechanical performance and difficulty controlling the loading and release of hydrophilic drugs onto/from the polymeric matrices.

Various nanoparticles have been used as drug carriers [20,21]. Among them, mesoporous silica nanoparticles, in particular Mobil Composition of Matter N0.41 (MCM-41), have been extensively exploited in a variety of bio-applications, including drug/gene delivery. MCM-41 consists structurally of hexagonally shaped pores with diameters ranging between 2 and 50 nm and a large surface area (900–1500 m$^2$ g$^{-1}$) [22]. The wall structure of pores of MCM-41 comprises a high density of disordered network of siloxane bridges and free silanol groups. Its external surface silanol groups (-Si-OH) can be functionalized and tailored in order to increase drug loading capacity and drug release efficiency of

the modified nanoparticles. Besides this, the chemical composition of MCM-41 is rather analogous to those of bioactive glasses that have been recently used in bone-specific drug delivery systems [23,24].

Recently, MCM-41 has been incorporated in polymeric biomaterials for small-molecule drug delivery applications. For instances, Hassan *et al.* [25] have produced antibiotic loaded carboxymethylcellulose/MCM-41 nanocomposite hydrogel films using citric acid as a crosslinker. The antibiotic release profiles and antibacterial activities of the films were suitable for wound dressing applications. Chunli *et al.* [26] demonstrated that the carrier prepared from the surface-modified MCM-41 nanoparticles conjugated with CMCS could enhance the loading of a pesticide onto it and provide a better bioactivity against the fungus, compared to the unmodified MCM-41.

To the best of our knowledge, a preparation of composite hydrogels consisting of CMCS and drug-loaded MCM-41, which had dual antibacterial activity and osteogenic potency, has rarely been studied and reported. In the present study, we successfully developed CDM-loaded MCM-41-CMCS hydrogels with combined controlled release of CDM and osteogenic potency for treatments of intraoral bone defects for the first time. CDM was initially loaded into the surface-modified MCM-41 nanoparticles, followed by the incorporation of CDM-loaded MCM-41 into CMCS. The resulting composite hydrogels were later exposed to a heat treatment. The effects of the surface modification of MCM-41 and the concentration of CDM loaded on the properties of the composite hydrogels, in terms of morphology, CDM loading and releasing capacity, and stability, were thoroughly investigated. Furthermore, the antibacterial and *in vitro* osteogenic activities for at least 10 days of measurement were assessed.

# 2. Experimental procedures

## 2.1. Materials

Water-soluble CMCS ($\bar{M}_w = 3.0 \times 10^5$ Da, degree of substitution (DS) = 0.9) was directly prepared in our laboratory, according to the method described in the literature [27]. MCM-41 mesoporous silica nanoparticles were synthesized according to our previous report [28]. CDM hydrochloride (MW = 479.46 g mol$^{-1}$) was supplied by Sigma-Aldrich Corporation. Absolute ethanol (EtOH) and acetonitrile (ACN) (HPLC grade, 99.9%) were purchased from CT chemical Co., Ltd (Thailand). All reagent-grade chemicals were used as received without further purification.

## 2.2. Surface modification of MCM-41 by oxygen plasma treatment

To enhance the surface hydrophilicity of MCM-41, the nanoparticles were treated with a low-pressure oxygen plasma treatment using an oxygen plasma reactor (model PDC-002, Harrick), where the chamber was evacuated to a total pressure of 205 mTorr. Electrodes were interconnected with a 13.56 MHz radio frequency (RF) generator. The oxygen plasma was ignited at 30 W, and the MCM-41 nanoparticles were subjected to the plasma treatment for 1 h, yielding the plasma-treated MCM-41 nanoparticles (coded as p-MCM-41). Both MCM-41 and p-MCM-41 nanoparticles (coded as (p)-MCM-41) were comparatively characterized by BET surface area analysis, X-ray diffraction analysis (XRD) and X-ray photoelectron spectroscopy (XPS).

## 2.3. Loading of CDM into (p)-MCM-41 nanoparticles

CDM was, in this study, used as a model drug to be loaded into the mesoporous silica nanoparticles. The adsorption of CDM into (p)-MCM-41 nanoparticles is schematically displayed in figure 1*a*. Typically, 50 or 200 mg of CDM powder was dissolved in 3 ml of de-ionized water (DI), and 420 mg of dried MCM-41 or p-MCM-41 was then added into the CDM solution and continuously stirred at room temperature for 24 h. The mixture was finally centrifuged at 6000 r.p.m. for 10 min, and the solid particles were collected and freeze-dried. The CDM-loaded mesoporous silica nanoparticles, generally referred to as (p)-MCM-41-CDM, were specifically assigned based on the type of MCM-41 and amount of CDM initially used, e.g. MCM-41-50CDM prepared by soaking MCM-41 in the aqueous solution of 50 mg of CDM.

(a)

(b)

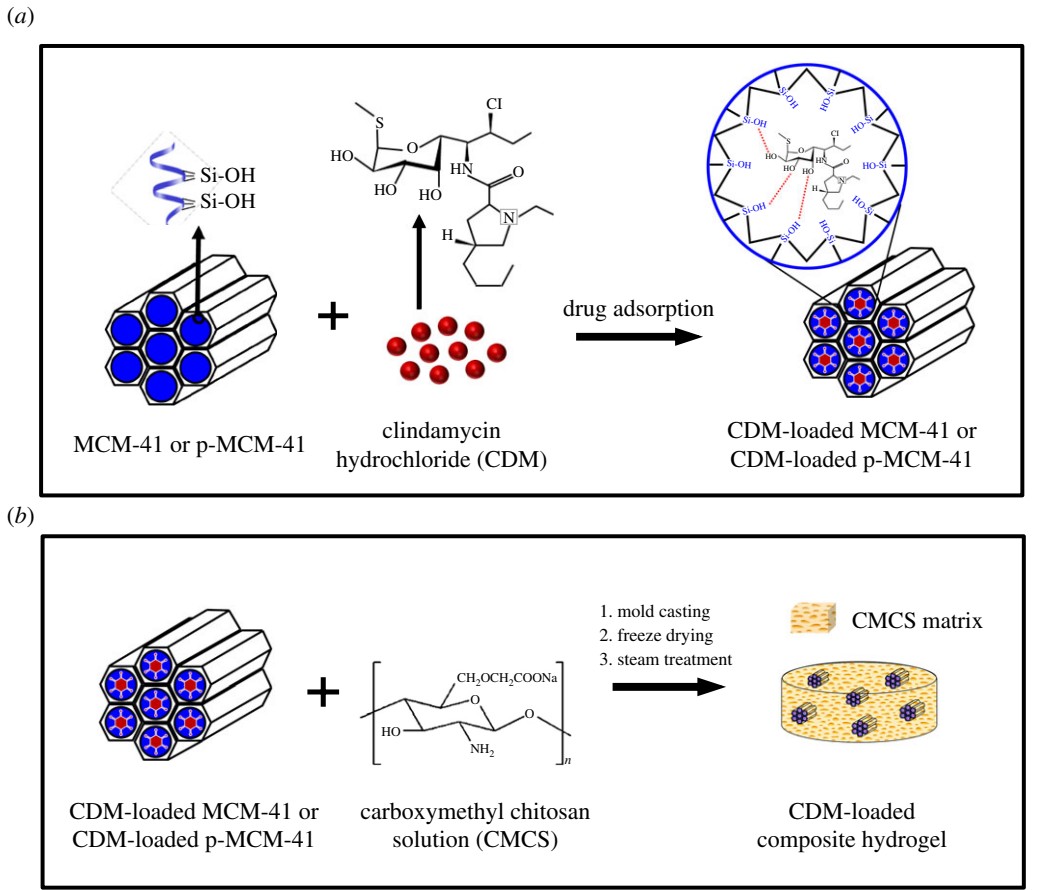

**Figure 1.** Schematic illustration of preparations of: (a) CDM-loaded MCM-41 nanoparticles and (b) MCM-41-CDM-CMCS composite hydrogels.

## 2.4. Characterization of (p)-MCM-41 and CDM-loaded MCM-41 nanoparticles

The specific surface area and pore characteristics of (p)-MCM-41 were measured from the nitrogen adsorption-desorption isotherms using a Brunauer–Emmett–Teller (BET) surface area analyzer (Quantachrome Instruments, Autosorb-1C). Powder X-ray diffraction (XRD), carried out using a Rigaku X-ray diffractometer (PANalytical XPert PRO, Jade software 9.7), was employed to confirm the structural characteristics of MCM-41, p-MCM-41 and CDM-loaded mesoporous materials. The diffraction patterns were collected in the range of $2\theta = 0.5$–$5°$ using a scanning rate of $0.01°/s$. The elemental compositions and surface chemical properties of MCM-41 nanoparticles before and after the plasma treatment were comparatively analysed by X-ray photoelectron spectroscopy (XPS) on a Kratos AXIS Supra™ electron spectrometer using a monochromatic Al K$\alpha$ 1,2 radiation with photon energy of 1.4 keV.

## 2.5. Preparation of CDM-loaded hydrogels

The preparation of CDM-loaded composite hydrogels is schematically shown in figure 1b. In brief, 630 mg of CMCS powder was primarily dissolved in 7 ml of DI water at room temperature, whereas 420 mg of CDM-loaded MCM-41 or CDM-loaded p-MCM-41 prepared in §2.3 was dispersed in 3 ml of DI water before being added into the CMCS solution. The fixed mass ratio of CMCS to CDM-loaded mesoporous material equal to 60 : 40 was used throughout this study. The individual viscous mixtures were continuously stirred for 5 h prior to being poured into Teflon moulds ($10 \times 7 \times 0.2$ cm$^3$) and lyophilized to produce sponge-like pads. Afterward, the CMCS matrix was crosslinked by exposing the porous pads to a steam treatment at 105°C for 15 min [27]. After drying in a high-pressure vacuum oven, the crosslinked pads were cut into small disks with a diameter of 4 mm and a thickness of 2 mm. The prepared composite hydrogel samples, generally referred to as (p)-MCM-41-CDM-CMCS, were specifically coded based on the type of CDM-loaded materials used, e.g. MCM-41-50CDM-CMCS prepared from the mixture of 40 wt% MCM-41-50CDM and 60 wt% CMCS.

The CMCS hydrogel (using 10.5% (w/v) viscous aqueous solution of 1050 mg CMCS in 10 ml DI water), 200CDM-CMCS hydrogel (using 12.5% (w/v) viscous aqueous solution of 1050 mg CMCS and 200 mg CDM in 10 ml DI water), and MCM-41-CMCS hydrogel (using viscous mixture of 630 mg CMCS and 420 mg MCM-41 in 10 ml DI water) were prepared using the same procedure as mentioned above.

## 2.6. Characterizations of CDM-loaded hydrogels

Surface morphology and pore size of the prepared composite hydrogel were examined in comparison with those of the porous CMCS hydrogel using a scanning electron microscope (SEM) (Hitachi, S-3400N) at an accelerating voltage of 10 kV after coating the specimens with gold in a sputtering device. The average pore size of each hydrogel was calculated directly from its SEM image using Image J/Fuji software, using at least 50 pores ($n = 50$). Thermal gravimetric analysis-differential thermal analysis (TGA-DTA) (Mettler Toledo, TGA/SDTA 851e) was conducted under $N_2$ and $O_2$ atmospheres to determine the CDM loading capacity in the nanoparticles. The TGA-DTA data were collected over the temperature range of 30–800°C with switching from $N_2$ to $O_2$ at 600°C at a heating rate of 20°C min$^{-1}$. The chemical structures of CDM, CMCS and CDM-loaded CMCS hydrogel were also comparatively analysed using Fourier-transform infrared spectroscopy in the attenuated total reflectance (ATR-FTIR) mode (Nicolet 6700, Thermo Fisher Scientific Inc., Madison, WI, USA). The ATR-FTIR spectra were collected in the range of 4000–600 cm$^{-1}$ with 32 scans.

## 2.7. In vitro drug releases of CDM-loaded hydrogels

The amounts of CDM released from (p)-MCM-41-CDM powder and (p)-MCM-41-CDM-CMCS hydrogels were analysed by high performance liquid chromatography (HPLC; WATERS HPLC 2965 SYSTEM). Briefly, about 20 mg of (p)-MCM-41-CDM powder or about 3.3 mg of (p)-MCM-41-CDM-CMCS composite hydrogel had been immersed in 1.2 ml of phosphate buffer saline (PBS) for 10 days at 37°C. A 1 ml aliquot of supernatant was collected daily for HPLC analysis, to measure the amount of CDM released; after liquid collection, 1 ml of fresh PBS was subsequently added to maintain the 1.2 ml volume of the test specimen. The HPLC mobile phase used was the mixed solvent of 0.02 M disodium hydrogen phosphate ($Na_2HPO_4$) (pH = 2.5) and acetonitrile, prepared at a volume ratio of 70 : 30. The flow rate and detection wavelength used were 0.5 ml min$^{-1}$ and 200 nm, respectively. The aliquots were individually filtered using 0.45 µm PTFE membranes before being injected at the volume of 100 µl. The reading amount of CDM in each aliquot was determined by correlating a CDM peak intensity in an HPLC chromatogram to the standard curve. The total amount of drug released per weight of (p)-MCM-4 powder ($R_{CDM}$) (%) was calculated using the equation below.

$$R_{CDM}\ (\%) = \frac{R_T}{M_R} \times 100, \tag{2.1}$$

where $R_T$ and $M_R$ represent the amount of total CDM released ($\sum_{t=1}^{t=10} R_t$) measured by HPLC and dry weight of (p)-MCM-41-CDM powder before release.

The amount of CDM remaining (%) in (p)-MCM-41-CDM powder after 10-day release was calculated using the equation below.

$$\text{drug remaining}\ (\%) = (A_{CDM} - R_{CDM}) \times 100, \tag{2.2}$$

where $A_{CDM}$ was the total amount of drug actually loaded per weight of (p)-MCM-4 powder before release, determined by TGA-DTA (the calculation detail is shown in electronic supplementary material 3).

## 2.8. In vitro stability of CDM-loaded hydrogels

The stability of (p)-MCM-41-CDM-CMCS hydrogels in a cell culture medium was assessed. In brief, small disks, each of (p)-MCM-41-CDM-CMCS hydrogel, prepared in §2.5, were individually placed in culture wells containing 1.2 ml of the cell culture medium (α-minimum essential medium, α-MEM) (pH 7.4) containing 10% fetal bovine serum (FBS), 200 U ml$^{-1}$ penicillin and 200 µg ml$^{-1}$ streptomycin (all from Gibco Life Technologies Ltd, Paisley, UK), and incubated at 37°C under 5% $CO_2$ atmosphere. The stability of the hydrogel specimens was evaluated daily up to 14 days with the medium replacement in every 2 days. On each incubation day, the remaining specimen was gently washed with DI water, freeze-dried overnight and eventually weighed. The remaining weight of the hydrogel

(%) on each day was calculated using the equation below.

$$\text{remaining weight (\%)} = \frac{M_a}{M_b} \times 100, \quad (2.3)$$

where $M_b$ and $M_a$ represent the weights of hydrogel before and after being incubated in the culture medium on a given incubation day, respectively ($n = 3$).

## 2.9. *In vitro* antibacterial activity of CDM-loaded composite hydrogels

### 2.9.1. *In vitro* antibacterial activity assessment

*Streptococcus sanguinis* (ATCC® 10 556™) suspension was incubated overnight, diluted with brain heart infusion (BHI) broth, and then adjusted to an equivalent to 0.5 McFarland standard (at an optical density in the range of 0.08–0.10 at 600 nm) before use. Meanwhile, a small disk of each (p)-MCM-41-CDM-CMCS hydrogel, prepared in §2.5, was sterilized under ultraviolet light for 1 h per each side and then placed in a 1.7 ml microcentrifuge tube. To the tube, 1.2 ml of the diluted inoculum of $3 \times 10^5$ to $7 \times 10^5$ CFU ml$^{-1}$ was added, and the whole tube was incubated at 37°C for 24 h. The tested sample was successively transferred to a new tube containing a fresh 1.2 ml diluted inoculum after 24 h incubation, up to 14 days. To evaluate the antibacterial efficiency of each (p)-MCM-41-CDM-CMCS hydrogel on a given incubation day, each bacterial solution incubated with a specimen (100 µl) was spread over a BHI agar plate and allowed to dry before being incubated at 37°C for 48 h. The antibacterial efficiency (%) was determined by using the equation below.

$$\text{antibacterial efficiency (\%)} = \frac{\text{CFUcontrol} - \text{CFUsample}}{\text{CFUcontrol}} \times 100, \quad (2.4)$$

where $\text{CFU}_{\text{sample}}$ and $\text{CFU}_{\text{control}}$ are the colony-forming units found in the bacterial solution with and without the test specimen, respectively ($n = 2$).

Bacteriostatic and bactericidal activities were defined as less than 3-log$_{10}$ and ≥3-log$_{10}$ (99.9%) reductions in CFU/ml at 24 h, respectively, compared with the starting inoculum [29].

### 2.9.2. Bacterial morphology observation

The morphological and structural features of *Streptococcus sanguinis* were observed by SEM after being incubated with each CDM-loaded composite hydrogel for given periods. A CDM-free composite hydrogel was used as a control. The whole specimens were rinsed twice with PBS and then fixed with 2.5% (w/v) glutaraldehyde at 4°C. After overnight incubation, the samples were individually rinsed with DI water and subsequently dehydrated for 10 min with series of ethanol solutions at 10%, 30%, 50%, 70%, 90% and 100% (twice in each step). The dehydrated specimens were dried by a critical point drying (CPD) technique and sputter-coated with gold before SEM analysis.

## 2.10. *In vitro* osteogenic activity of the CDM-loaded composite hydrogel

### 2.10.1. Cell culture

Human mesenchymal stem cells (hMSCs; Lonza Biologics plc, Cambridge, UK) were maintained in a standard medium α-MEM containing 10% FBS supplemented with 200 U ml$^{-1}$ penicillin, 200 µg ml$^{-1}$ streptomycin and 2 mM L-glutamine (all from Gibco) at 37°C under 5% CO$_2$ atmosphere. All specimens were UV sterilized for 1 h and pre-incubated in the culture medium for 24 h before all experiments described below.

For the cell viability test, hMSCs were seeded at a density of $5 \times 10^3$ cells cm$^{-2}$ in 24-well plates and cultured in the medium for 1, 3 and 7 days with the medium being changed every 48 h. For osteogenic differentiation, hMSCs were plated at a density of $2 \times 10^4$ cells cm$^{-2}$ in 24-well plates and allowed to grow in the medium for the first 48 h until the cells reached 80% confluence. Then, the cells were incubated with or without an osteogenic medium (OM) (standard medium with 100 nM dexamethasone, 50 µM ascorbate-phosphate and 10 mM β-glycerolphosphate; all from Sigma). The cells were cultured for 3 days (for alkaline phosphatase (ALP) activity assay) and 14 days (for gene expression and mineralization assays). The CDM-loaded composite hydrogel sample was placed in culture by using a semi-permeable porous membrane (0.4 µm) cell culture insert (Nunc, VWR Ltd, Lutterworth, UK). A schematic diagram of the culture with the hydrogel is shown in figure 6a. For all experiments with the

CDM-loaded composite hydrogel, the volume of culture medium per well was 1.2 ml, with the medium being refreshed every 48 h. The cultures without the CDM-loaded composite hydrogel were used as controls.

### 2.10.2. Cell viability assay

After 1, 3 and 7 days in culture, 3-(4,5-dimethylthiazol-2-yl)-2,5-diphenyltetrazolium bromide (MTT) assay was used to examine the viability of hMSCs. After being cultured at the indicated times, the cells were incubated with 0.2% MTT solution at 37°C for 4 h, and the reaction was then stopped by adding dimethyl sulfoxide (DMSO) and glycine buffer. The end product colour was subsequently analysed by measuring an absorbance at 490 nm ($A_{490}$) which corresponds to the viability of cells.

### 2.10.3. RNA extraction and reverse transcription (RT) quantitative real-time polymerase chain reaction (QPCR)

After 14 days in culture, total RNA was extracted from the cells using RNeasy® Mini Kit (Qiagen, West Sussex, UK), in accordance with manufacturer's instructions. To assess the quality of the extracted RNA and as an internal RT-PCR standard, the housekeeping gene glyceraldehyde 3-phosphate dehydrogenase (GAPDH) was also amplified. For reverse-transcription reaction, 1 µg of total RNA was used with 5 ng oligo-dT (Promega, Madison, WI) in 40 µl of water. After 5 min at 65°C, the first stand of cDNA was synthesized in a total volume of 50 µl, containing 50 U of cloned Moloney murine leukemia virus (M-MuLV) reverse transcriptase, 1× M-MuLV buffer, 40 µM of each dNTP and 40 U of RNase block (all Stratagene, La Jolla, CA). After incubation at 37°C for 60 min, the enzyme was inactivated by incubation at 90°C for 5 min, after which 1 µl of each cDNA sample was used in Q-PCR, as described below.

The first strand cDNA was subjected to Q-PCR using SYBR Green I dye performed in an iQ5 iCycler (BioRAd, Bradford, UK), with specific primers for the runt-related transcription factor 2 (RUNX2), type-I collagen (COL-I), osteocalcin (OCN) and GAPDH mRNA. GAPDH was used as an endogenous control. SYBR Green PCR reaction mixtures using SYBR Green I Master kit (Roche Diagnostic Co.) were set up as suggested by the manufacturer. All PCR reactions were performed in six replicates, and each of the signals was normalized to the GAPDH signal in the same reaction. The mRNA expression is expressed as mean fold-change of control (1.0). Data are presented as the mean fold-change ± s.d. from three independent experiments. Primer sequences were as follows: RUNX2 F 5′-TGGTTACTGTCATGGCGGGTA-3′, R 5′-TCTCAGATCGTTGAACCTTGCTA-3′; COL-I F 5′-GAGGGCCAAGACGAAGACATC-3′, R 5′-CAGAT CACGTCATCGCACAAC-3′; OCN F 5′- CACTCCTCGCCCTATTGGC-3′, R 5′-CCCTCCTGCTTGGAC ACAAAG-3′; GAPDH F 5′- CTGGGCTACACTGAGCACC-3′, R 5′- AAGTGGTCGTTGAGGGCAATG-3′ [30,31].

### 2.10.4. ALP activity assay

After 3 days in culture, the ALP activity of hMSCs was determined. Cells were analysed by RIPA lysis buffer. Cell lysates were collected and centrifuged at 2500 r.p.m. for 30 min, and the supernatants were obtained for analysis. The protein contents in the aliquots of supernatant were measured using the BCA protein kit (Pierce, Rockford, IL). The aliquots with an equal amount of protein were analysed for the ALP activity by addition of 10 mM *p*-nitrophenylphosphate (*p*-NPP; Sigma) and incubation at 37°C for 10 min. The reaction was then stopped by 3 N NaOH. The production of *p*-nitrophenol (*p*-NP) was measured spectroscopically at $A_{405}$.

In some experiments, before exposure of hMSCs to the CDM-loaded composite hydrogel sample, cells were pre-treated (2 h) with 1 µM U0126, 10 µM SP600125 and 10 µM SB203580, which are inhibitors specific to extracellular regulated kinase (ERK) 1/2, Jun N-terminal kinase (JNK) 1/2/3 and p38 mitogen activated protein kinase (MAPK), respectively.

### 2.10.5. Mineralization assay

Mineralization was quantified by an alizarin red S assay. After 14 days in culture, the cells were fixed with cold methanol at 4°C for 30 min and washed with distilled water. Subsequently, 1% alizarin red S (pH 4.2; Sigma) was added onto the samples and incubated at room temperature for 10 min, and the samples were rinsed twice with methanol to remove unbound alizarin red S. Incorporated alizarin red S was extracted from the matrix by the addition of 100 mmol $l^{-1}$ cetylpyridinium chloride (Sigma-Aldrich) at room temperature for 20 min, and the absorbance ($A_{570}$) was measured. The absorbance

value of each sample was obtained from the subtraction of the absorbance of a blank sample without cells from that of a sample with cells.

## 2.11. Statistical analysis

The data were presented as the mean ± s.d. based on three independent experiments, with the experiments being performed in at least triplicate. Statistical differences were analysed by means of SPSS software (SPSS, Inc., Chicago, IL) using a one-way ANOVA followed by the post hoc Dunnett's test with $p$-value $< 0.05$ being considered statistically significant.

# 3. Results

## 3.1. Properties of CDM-loaded untreated- and plasma-treated MCM-41 nanoparticles

The BET specific surface area, total mesopore volume, and average pore diameter of MCM-41 were measured using the nitrogen ($N_2$) adsorption/desorption isotherms. The $N_2$ adsorption/desorption isotherms obtained were type IV isotherms (see electronic supplementary material, figure S1), according to the IUPAC classification [32]. The determined surface area of MCM-41 was 1,726 $m^2 \, g^{-1}$. The total mesopore volume and the average pore diameter calculated from the isotherms were 1.33 $cm^3 \, g^{-1}$ and 26.6 Å, respectively, indicating a highly ordered mesoporous network with a hexagonal array. Moreover, the well-defined steps at low relative pressure $p/p_0$ of 0.5–0.9 suggested uniform mesoporous channels and narrow pore distribution (displayed in the inset of electronic supplementary material, figure S1).

The elemental compositions and framework bonding nature of the unmodified MCM-41 and plasma-treated MCM-41 nanoparticles were comparatively determined by XPS. The XPS survey scan spectra indicated that carbon (C 1 s at 285 eV), oxygen (O 1 s at 532 eV) and silicon (Si 2p at 103.8 eV) were the main elements on the surfaces of both types of nanoparticles; their surface element concentrations (%) were comparatively tabulated in electronic supplementary material, table S1. The enhanced O 1 s concentration in p-MCM-41 was partly owing to the plasma ion etching of the material surface, newly generating polar functional groups (some activated oxygen or hydroxyl groups) on the nanoparticle surface [33]. Moreover, the oxidation of the silanol groups on the surface of MCM-41 resulted in an increase in the oxygen concentration. Consequently, the calculated O/Si atomic ratio (%) of p-MCM-4 ($3.42 \pm 0.19$) was higher than that of MCM-41 ($2.44 \pm 0.91$). This was in accordance with the results reported on the surface modification of amorphous $SiO_2$ nanoparticles by oxygen-plasma treatment [34].

The low angle XRD patterns demonstrated in electronic supplementary material, figure S2 revealed the crystalline phase of MCM-41, p-MCM-41, and CDM-loaded p-MCM-41 nanoparticles. The XRD pattern of MCM-41 exhibited a main (100) peak at $2\theta = 2.31°$ (JCPDS no. 00-049-1712) [35], reflecting the ordered two-dimensional hexagonal structure. An additional board peak at $2\theta$ value of 4.2° suggested the formation of disordered phases. After plasma treatment, the intensity of the peak at 2.31° was considerably decreased due to an increase in the polar components on the material surface. In CDM-loaded p-MCM-41, the intensity of the band at 2.31° was further subsided, resulting from the increased electron density within the CDM-filled pores. A similar finding was reported by Heikkilä *et al.* [33], who illustrated the effects of the different mesoporous silica carriers on the drug uptake capacity. However, all three nanoparticles possessed similar crystalline phase patterns, implying negligible effects on the disordered mesoporous phase.

## 3.2. Hydrogel morphology

The cross-sectional morphology of a CDM-free composite hydrogel, i.e. MCM-41-CMCS, was examined in comparison with that of CMCS hydrogel by SEM. The SEM images in figure 2 demonstrate a denser porous structure of the composite hydrogel, owing to the presence of the agglomerated inorganic nanoparticles (MCM-41), as seen in the magnified image in figure 2d. The pore surface of the CMCS hydrogel appeared far smoother. The average pore sizes of the CMCS and composite hydrogels, measured from their SEM images using ImageJ, were 145 ± 31 and 98 ± 21 µm, respectively.

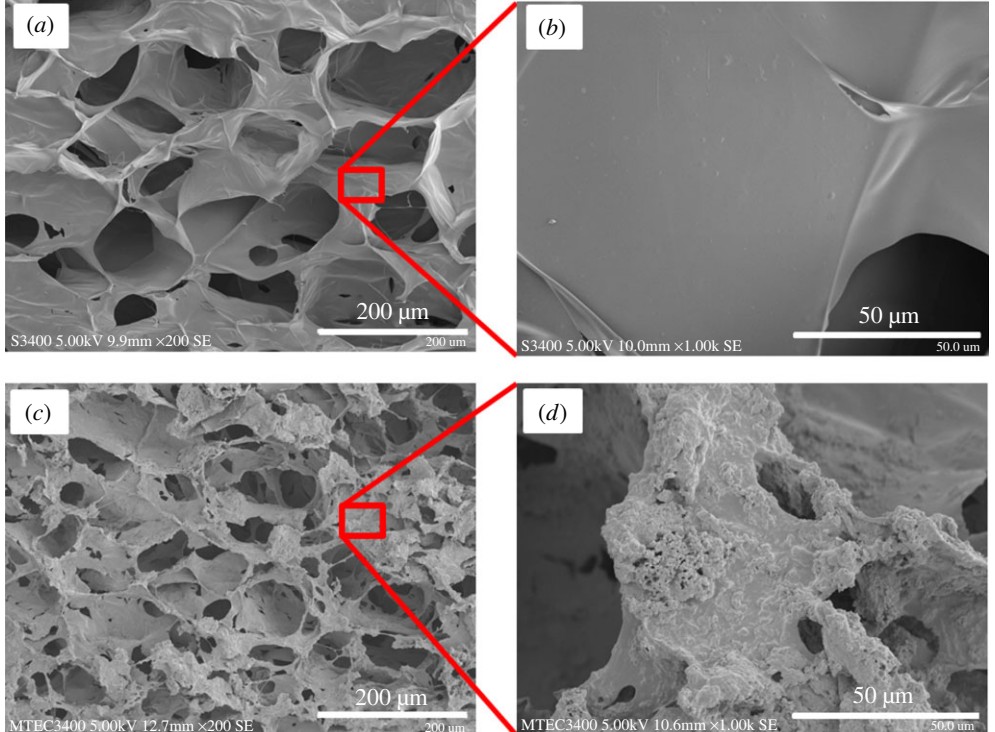

**Figure 2.** The cross-sectional SEM images of the porous hydrogels: (a,b) CMCS and (c,d) MCM-41-CMCS (left images; bar = 200 μm and right images; bar = 50 μm).

### 3.3. *In vitro* CDM loading and release

The total amounts of CDM actually loaded ($A_{CDM}$) per weight of (p)-MCM-4 powder before release were determined from the TGA-DTA thermograms of CDM, (p)-MCM-41 and (p)-MCM-41-CDM using the % remaining weights of individual samples at 500–800°C, as shown in electronic supplementary material, figure S3 with the calculation detail in electronic supplementary material, S3. Table 1 shows the percentages of total CDM adsorbed, released, and remaining per weight of specimen found in individual CDM-loaded (p)-MCM-41. Obviously, the plasma treatment of the nanoparticles prior to the drug loading considerably intensified the CDM adsorption. The enhanced surface hydrophilicity of MCM-41 explicitly boosted the CDM loading capability.

The concentrations of drug release (μg ml$^{-1}$) versus time of each (p)-MCM-41-CDM or (p)-MCM-41-CDM-CMCS hydrogel determined by high performance liquid chromatography (HPLC) were tabulated in table 2. The burst CDM releases from all samples were observed on day 1 of measurement, followed by slow and nearly flattened releases of CDM up to 10 days. Not only the similar release profiles, but also the nearly comparable CDM concentrations found on each day, were surprisingly noted in the (p)-MCM-41-CDM-CMCS hydrogels prepared from the same content of CDM initially loaded. Overall, the total amounts (μg) of CDM released from both MCM-41-200CDM-CMCS and p-MCM-41-200CDM-CMCS were nearly equal. Likewise, the similar release profiles of (p)-MCM-41-CDM were perceived. Consequently, greater contents of CDM still remained in the plasma-treated nanoparticles and hydrogels. It was noteworthy that a prolonged release (up to 10 days) was still satisfactorily achieved in the composite hydrogel mixed with p-MCM-41-50CDM.

### 3.4. Hydrogel stability

The physical stability of two p-MCM-41-CDM-CMCS hydrogels was evaluated in comparison with that of CDM-free p-MCM-41-CMCS hydrogel in a culture medium by measuring the remaining weight of each specimen on each day up to 14 days, using a gravimetrical method. As illustrated in figure 3, in the absence of CDM, the crosslinked p-MCM-41-CMCS hydrogel specimen absolutely disappeared in the medium within 4 days, but the CDM-loaded composite specimens still remained stable on day 14 with the weight losses approximately only 30%. Increasing CDM initially loaded into the composite

**Table 1.** The total amounts of drug actually loaded, released and remaining per weight of CDM-loaded nanoparticle sample.

| sample | $A_{CDM}$ (%) | $R_{CDM}$ (%) | drug remaining (%) |
|---|---|---|---|
| MCM-41-200CDM | 5.17 ± 1.54 | 3.34 ± 0.20 | 1.83 ± 0.87 |
| p-MCM-41-200CDM | 29.77 ± 2.57 | 3.86 ± 0.14 | 25.91 ± 1.35 |
| p-MCM-41-50CDM | 14.58 ± 1.04 | 1.00 ± 0.09 | 13.64 ± 0.99 |

**Table 2.** CDM release profiles and total amounts of CDM released from each CDM-loaded sample.

| day | CDM concentration ($\mu g\ ml^{-1}$) | | | | |
|---|---|---|---|---|---|
| | MCM-41-200CDM | p-MCM-41-200CDM | MCM-41-200CDM-CMCS | p-MCM-41-200CDM-CMCS | p-MCM-41-50CDM-CMCS |
| 1 | 486.7 ± 29.3 | 581.8 ± 46.7 | 220.1 ± 8.2 | 243.5 ± 11.6 | 74.5 ± 6.7 |
| 2 | 47.9 ± 2.6 | 56.9 ± 0.1 | 19.2 ± 0.4 | 23.7 ± 0.8 | 8.1 ± 0.1 |
| 3 | 8.4 ± 1.1 | 11.6 ± 0.2 | 4.1 ± 0.2 | 4.1 ± 0.1 | 1.9 ± 0.3 |
| 4 | 2.9 ± 0.2 | 6.1 ± 0.3 | 2.2 ± 0.1 | 1.9 ± 0.1 | 1.2 ± 0.1 |
| 5 | 1.7 ± 0.2 | 2.5 ± 0.1 | 1.6 ± 0.1 | 1.6 ± 0.1 | 0.8 ± 0.1 |
| 6 | 1.6 ± 0.2 | 2.2 ± 0.2 | 1.2 ± 0.1 | 1.5 ± 0.1 | 0.7 ± 0.1 |
| 7 | 1.1 ± 0.1 | 1.0 ± 0.1 | 1.2 ± 0.1 | 1.3 ± 0.1 | 0.7 ± 0.1 |
| 8 | 1.2 ± 0.1 | 0.7 ± 0.1 | 1.0 ± 0.1 | 0.8 ± 0.1 | 0.5 ± 0.1 |
| 9 | 0.9 ± 0.1 | 0.5 ± 0.1 | 0.7 ± 0.1 | 0.6 ± 0.1 | 0.5 ± 0.1 |
| 10 | 0.9 ± 0.1 | 0.4 ± 0.1 | 0.6 ± 0.1 | 0.5 ± 0.1 | 0.4 ± 0.1 |
| total amount of CDM released ($\mu g$) | 663.5 ± 39.5 | 768.9 ± 27.5 | 302.4 ± 9.9 | 334.4 ± 14.9 | 107.3 ± 6.3 |

hydrogel slightly improved the physical stability of p-MCM-41-200CDM-CMCS, compared to that of p-MCM-41-50CDM-CMCS.

To explore the effect of CDM on the stability of p-MCM-41-CDM-CMCS hydrogels, CDM-loaded CMCS hydrogel was used as a material model in the analysis of an interaction between the molecules of CDM and CMCS by FTIR-ATR. A specimen of CDM-loaded CMCS hydrogel was subjected to a 3-day drug release procedure as described in §2.7 in SI, to liberate excess CDM loosely bound to the hydrogel before evaluation because no CDM in the aliquot collected on day 4 was HPLC detected (data not shown). The overlaid FT-IR spectra of CDM, CMCS and 200CDM-CMCS samples are illustrated in figure 4. All characteristic absorption bands were observed as follows: 862 and 1685 cm$^{-1}$ corresponding to -C-Cl stretching and N–C=O stretching, respectively, in CDM [36]; and 1114–1170, 1594 and 1654 cm$^{-1}$ corresponding to bridge oxygen (C-O-C), -NH$_2$ deformation, and amide I band (-CONH-), respectively in CMCS [37]. The spectrum of 200CDM-CMCS sample obtained after 3 days of drug release clearly revealed combined major characteristic peaks of CDM and CMCS with a new tiny absorption band at 1740 cm$^{-1}$, corresponding to >C=O stretching of an ester bond, indicating an intermolecular crosslinking between hydroxyl groups (-OH) of CDM and carboxylate (-COO-) groups of CMCS (comparing to the spectrum of 200CDM-CMCS sample obtained before drug release as shown in electronic supplementary material, figure S4). Moreover, the broadened band at 1594 cm$^{-1}$ was caused by overlapped formations of amide linkages from the crosslinking between amine (-NH$_2$) and carboxylate groups in CMCS and an intermolecular crosslinking between -NH$_2$ in CDM and -COO- in CMCS. Hence, in (p)-MCM-41-CDM-CMCS, there existed the intermolecular crosslinking between CMCS and CDM physically adsorbed into (p)-MCM-41 nanoparticles. H-bonding formed

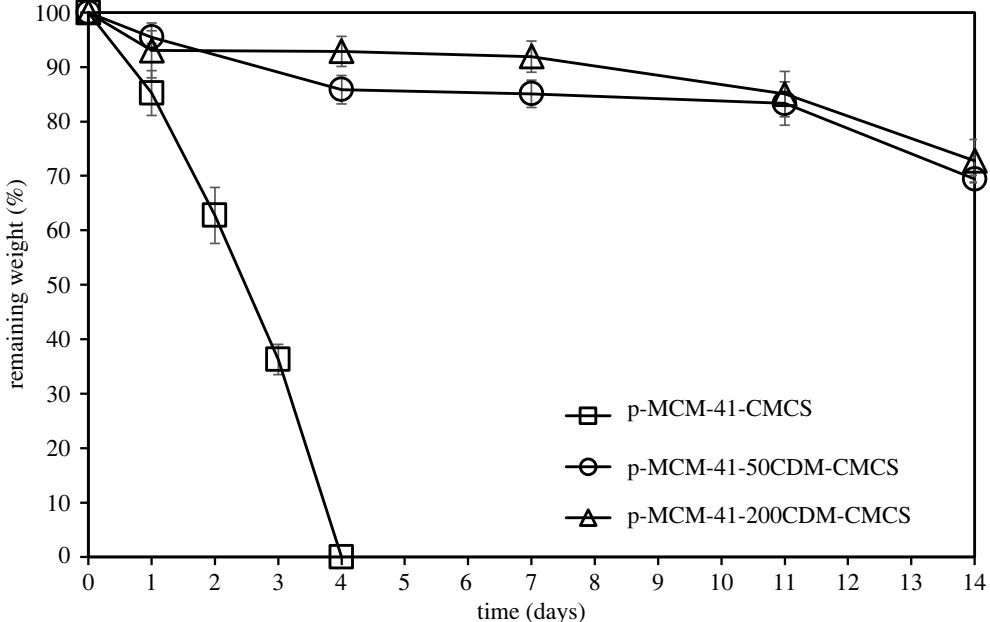

**Figure 3.** The remaining weight (%) of CDM-loaded composite hydrogels as a function of immersion time.

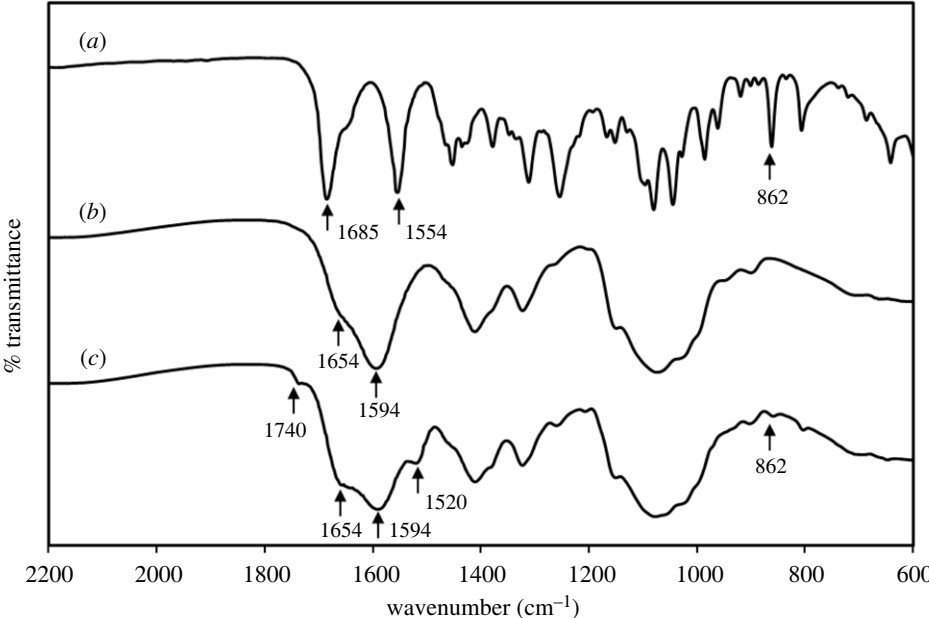

**Figure 4.** Overlaid FT-IR spectra of CDM, CMCS and CDM-loaded CMCS hydrogel.

between CDM and (p)-MCM-41 would exclusively help tie all substances together, resulting in the superior stability of the CDM-loaded composite hydrogels.

## 3.5. *In vitro* antibacterial activity of CDM-loaded composite hydrogels

CDM-susceptible gram-positive coccus specie *Streptococcus sanguinis* (ATCC® 10 556™) was exposed to three different CDM-loaded composite hydrogels. As revealed in figure 5a, the antibacterial activity of both MCM-41-200CDM-CMCS and p-MCM-41-200CDM-CMCS hydrogels slightly decreased from 100% on day 2 to approximately 95% on day 9. However, p-MCM-41-200CDM-CMCS possessed somewhat higher antibacterial activity than MCM-41-200CDM-CMCS between day 10 and day 14 (approximately 90% versus 83%). In marked contrast, the antibacterial activity of p-MCM-41-50CDM-CMCS appeared to be about 95% during day 2 to day 6 and sharply declined to an undetectable level

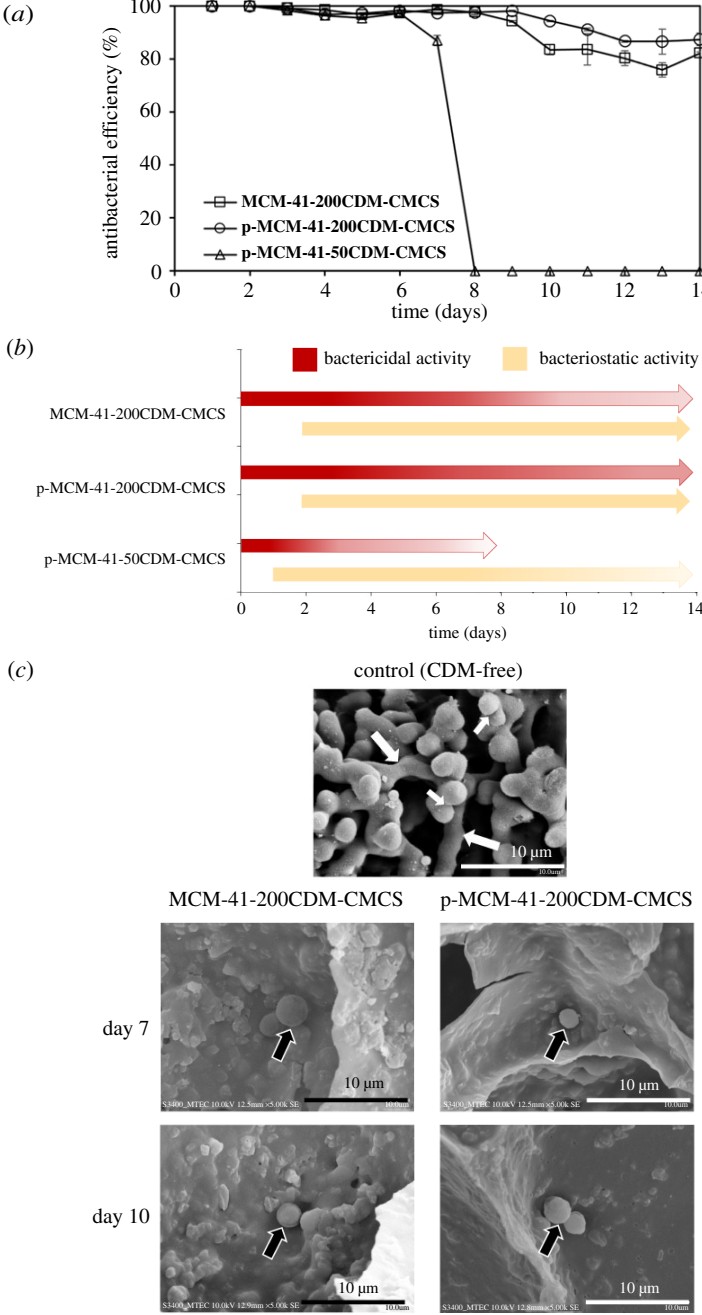

**Figure 5.** The antibacterial activity of ( p)-MCM-41-CDM-CMCS composite hydrogels against *Streptococcus sanguinis*. (a) The percentages of antibacterial efficiency of the composite hydrogels during 14-day incubation with bacterial suspension are comparatively shown. (b) Bactericidal and bacteriostatic activities of the composite hydrogels during 14-day incubation with bacterial suspension are summarized. Colour gradient of the block arrows display reductions in the effect on bacteria over time. (c) SEM micrographs show morphological features of bacteria exposed to the blank control composite hydrogel (without CDM loading), MCM-41-200CDM-CMCS and p-MCM-41-200CDM-CMCS for 24 h. Noted that *Streptococcus sanguinis* occurred in chains of cocci (large white arrows) and pairs of cocci (small white arrows) and the corrugated surface of only single or pair damaged cells (black arrows) are shown. Original magnification × 5000.

on day 8. The bacteriostatic activity of this composite hydrogel was observed during this duration and completely diminished after day 8. The MCM-41-200CDM-CMCS and p-MCM-41-200CDM-CMCS were positively able to kill all the bacteria in the first 48 h, indicating 100% bactericidal activity, whereas this was only observed in the first 24 h for p-MCM-41-50CDM-CMCS (figure 5b). Both 200CDM-loaded composite hydrogels possessed full bacteriostatic activity (i.e. the ability to inhibit bacterial growth) throughout the tested duration.

Figure 5c demonstrates SEM micrographs of *Streptococcus sanguinis* that occurred in chains of cocci (large white arrows) and pairs of cocci (small white arrows). The bacteria were densely aggregated on CDM-free composite hydrogel (used as a control) after 24 h of incubation. The SEM results further supported the antibacterial property of the CDM-loaded composite hydrogels. Morphological features of affected bacteria under SEM showed the corrugated surface of an individual single coccus cell. Only single or pair damaged cells were observed (black arrows). An enormously lower number of the bacteria were found on both MCM-41-200CDM-CMCS and p-MCM-41-200CDM-CMCS, compared with that observed on the control.

## 3.6. *In vitro* responses of CDM-loaded composite hydrogels to hMSCs

### 3.6.1. Cytotoxicity

At the early period in culture with the medium being changed every 2 days in order to simulate fluid exchange *in vivo*, approximately 95% viability of hMSCs exposed to p-MCM-41-200CDM-CMCS composite hydrogel, denoted as CDM hydrogel in figure 6, was detected, which was comparable to that of cells exposed to the control (without hydrogel), defined as 100% on day 1 (figure 6b). Moreover, hMSCs cultured with the hydrogel were able to proliferate with the viable cells increasing by approximately two- and three-fold in day 3 and day 7, respectively. On those days, the viability of hMSCs cultured with the CDM-loaded hydrogel remained similar to that observed in the control, suggesting that at all time points tested, the MCM-41-200CDM-CMCS hydrogel was not cytotoxic to hMSCs and did not affect the proliferation of hMSCs.

### 3.6.2. Mineralization

The effect of p-MCM-41-200CDM-CMCS composite hydrogel on *in vitro* mineralization was examined using hMSCs as progenitors of osteoblastic cells. Figure 6c demonstrates that cells unexposed to the composite hydrogel produced approximately 300% higher mineralization under osteogenic stimulation, compared with those cultured in standard medium, whereas more alizarin red-positive mineralized deposit was satisfactorily perceived in cells cultured with the hydrogel, compared with that found in the culture without hydrogel.

### 3.6.3. ALP activity and osteoblast differentiation

The mechanism by which the p-MCM-41-200CDM-CMCS hydrogel enhanced mineralization was investigated. We first examined the transcript expression of key genes associated with osteoblast differentiation by quantitative real-time PCR. Figure 6d shows that although, under osteogenic culture, the expression of RUNX2, COL-I and OCN mRNA was increased approximately two- to three-fold compared with that cultured under standard culture, the MCM-41-200CDM-CMCS composite hydrogel did not further upregulate the expression of any genes tested compared with the control cells. This suggests that enhanced mineralization by the CDM-loaded composite hydrogel might not involve the increased osteoblast differentiation of hMSCs.

Since an increase in cellular ALP activity has been associated with enhanced mineralization [38–40], the level of ALP activity in hMSCs cultured with the CDM-loaded composite hydrogel was determined. Figure 6e reveals that under osteogenic induction, hMSCs in the control well had approximately 170% ALP activity of cells in non-osteogenic stimulation, whereas the CDM-loaded composite hydrogel further induced the ALP activity of hMSCs under osteogenic culture to 265% of that in control cells. Moreover, the upregulated ALP activity induced by the material appeared to be mediated by p38 MAPK, but not ERK or JNK MAPKs, as seen by a significant reduction in the ALP activity only in the presence of the p38 MAPK inhibitor (figure 6f). The results suggest that the p-MCM-41-200CDM-CMCS composite hydrogel might induce hMSC-mediated mineralization via p38-dependent upregulated ALP activity.

# 4. Discussion

Hydrogels incorporated with drug carriers, such as nanoparticles, have attracted much attention for drug delivery systems in bone tissue engineering applications. Particularly, MCM-41 can be used effectively not only in adsorption and release of drug molecules, but in bone tissue engineering due to its chemical composition and surface structure with silicon dioxide based functional groups somewhat

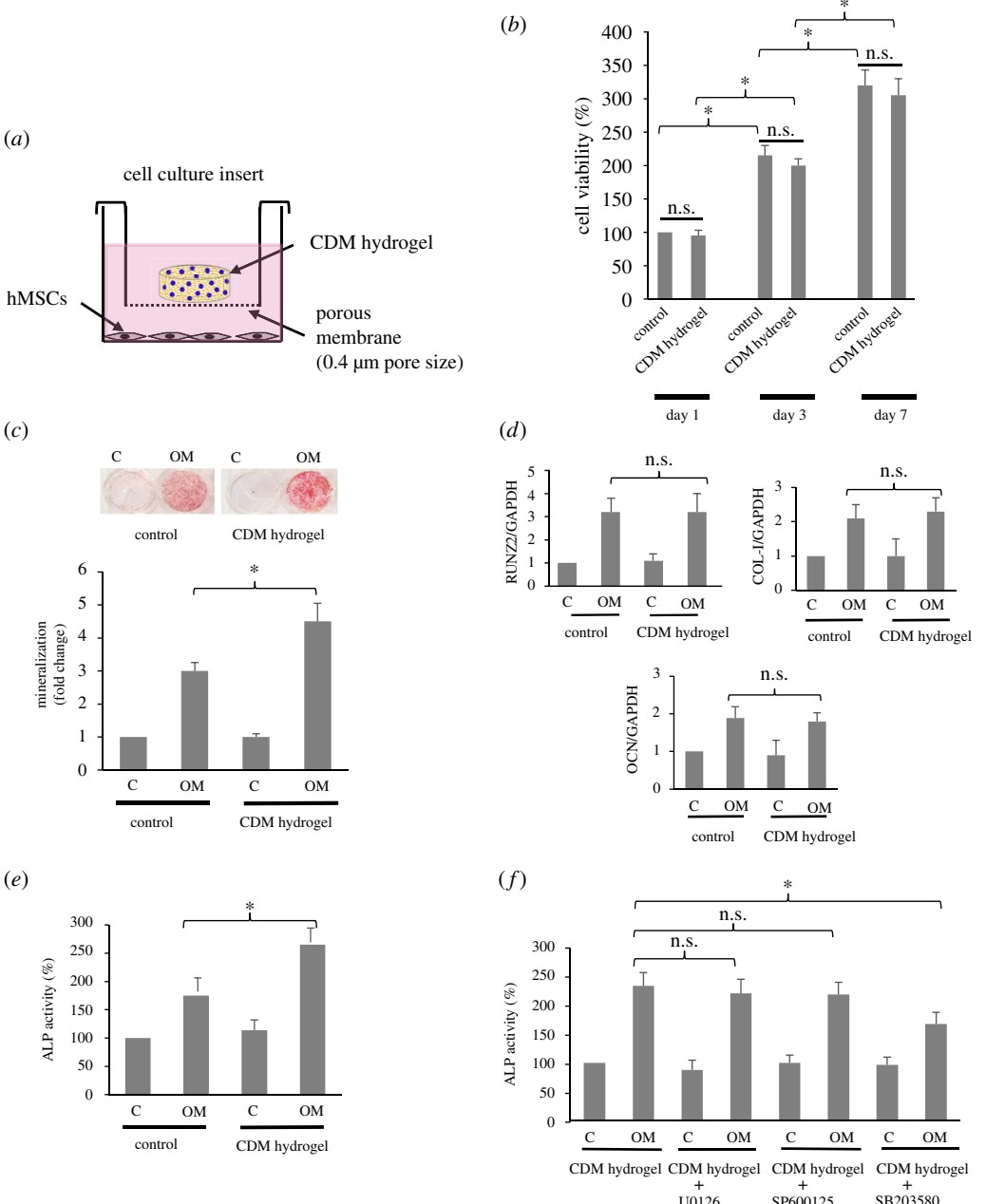

**Figure 6.** Osteogenic potency of the p-MCM-41-200CDM-CMCS composite hydrogel (CDM hydrogel). The experiment set up for the delivery of the hydrogel to hMSCs is depicted in (*a*). In (*b*), cells were cultured in a standard culture medium and exposed to the hydrogel and the cell viability was determined by MTT on days 1, 3, 7. Biomineralization determined by alizarin red assay (*c*), the expression of osteogenic genes RUNX2, COL-I and OCN assessed by Q-PCR (*d*) and the ALP activity determined by biochemical analysis (*e*) in hMSCs exposed to the hydrogel under osteogenic stimulation were also examined for its osteogenic potency. In (*f*), cells were pre-treated with chemical inhibitors U0126, SP600125 and SB203580 before exposure to the CDM hydrogel, and the ALP activity was determined. Cells cultured without the hydrogel or without osteogenic supplements were used as controls. The results are expressed as mean percentage (or fold change) ± s.d. of triplicates, defined as 100 (or 1) in the control group. Three independent experiments showed similar results. n.s., not significant; $^{*}p < 0.05$.

analogous to bioactive glasses [23]. In this work, (p)-MCM-41-CDM-CMCS composite hydrogels were successfully developed for treatment of jaw bone infection. The excellent CDM loading capacity and prolonged release profiles up to 10 days with full bacteriostatic activity up to 14 days of these composite hydrogels were evidently demonstrated. Furthermore, the p-MCM-41-200CDM-CMCS composite hydrogel appreciably promoted the osteogenic potency of MSCs. Approximately 30 wt% of p-MCM-41-200CDM-CMCS composite hydrogel gradually vanished in the culture medium within 14 days of culture; all the disintegrated substances possessed interesting biological properties that were

beneficial to hMSC-mediated biomineralization and curing of infected bones. For instance, MCM-41 was found to enhance osteogenic activity [41], and CDM exhibits a better bone penetration ability than commonly used antibacterial agents such as penicillin and metronidazole [4].

In this study, both MCM-41 and p-MCM-41 nanoparticles were used as CDM carriers. After $O_2$ plasma treatment, the formation of newly formed oxygen-containing polar functional groups, e.g. some activated oxygen and hydroxyl groups, on the surface of MCM-41 was observed by XPS, similar to the results previously reported [33,34]. The CDM loading into the plasma-treated nanoparticles was successfully conducted, confirmed by XRD analysis. As seen in electronic supplementary material, figure S2, there was no significant difference in d-spacing value of the peak (100) at 2.31° of the CDM-loaded p-MCM-41 nanoparticles, suggesting the structural stability of the nanoparticles [33]. However, the peak intensity was drastically reduced after CDM was incorporated into p-MCM-41.

In the preparation of (p)-MCM-41-CDM-CMCS composite hydrogel, CMCS polymer matrix was crosslinked by a steam treatment by following the method previously disclosed [27]. The crosslinking of CMCS proceeded via amidization of amino ($-NH_2$) and carboxylate ($-COO-$) groups, forming amide linkages. The steaming condition was primarily optimized to ensure that the stability of CDM was unperturbed. The HPLC analysis of aliquots collected on day 1 from p-MCM-41-200CDM-CMCS hydrogel steamed at varied temperatures, i.e. 100–121°C, for varied times, i.e. 10–15 min was carried out. In electronic supplementary material, figure S5, the HPLC chromatogram of original CDM demonstrated the major peak of CDM at a retention time of 7.2 min (marked with red arrow) with minor peaks of unknowns. Once CDM integrated in the composite hydrogels was exposed to the steam, the intensity of this major peak was lessened with increasing steaming temperature and time: $CDM_{121-15} < CDM_{115-15} < CDM_{105-10} \sim CDM_{100-15}$. Concurrently, the unknown peak at a retention time of about 5.2 min (marked with red arrows) oppositely intensified when harsh steaming conditions were exploited, implying the partial decomposition of CDM molecules upon heat treatment. As disclosed earlier, when CDM solution was heated at 100°C for 30 min, various degraded products and derivatives were generated due to the thermal degradation [42]. From these results, the crosslinking of (p)-MCM-41-CDM-CMCS composite hydrogels was essentially performed at 105°C for 10 min, to minimize the degradation of CDM and simultaneously obtain a sufficient degree of crosslinking of CMCS which directly influenced both physical stability and drug release profiles of the composite hydrogels.

The surface morphology and pore size of CDM-free composite hydrogel were initially investigated by SEM. The porous structure of the composite hydrogel appeared rather denser than that of the pure CMCS hydrogel; nonetheless, both types of CDM-free hydrogels still possessed interconnected pores with average sizes of $98 \pm 21$ and $145 \pm 31$ μm, respectively. The smaller pore size of the composite hydrogel was associated with the presence of the MCM-41 agglomerates on the matrix surface, as shown in figure 2. Nevertheless, the interconnected porous structure and pore size of approximately 100 μm could appropriately allow cells to infiltrate into the composite as well as good flows of oxygen, nutrients and fluids beneficial to the cells [43].

Generally, there are several influencing parameters on the loading and release of drugs into and from the materials, such as type of drug delivery materials, shape and structure of materials, and drug-material interaction [44]. The total quantities (%) of CDM truly loaded per weight of (p)-MCM-41 before release ($A_{CDM}$) determined from the TGA-DTA thermograms of CDM, (p)-MCM-41 and (p)-MCM-41-CDM using the %remaining weights of individual samples at 500–800°C are reported in table 1. Meanwhile, the amounts of drug released as a function of time of (p)-MCM-41-CDM analysed by HPLC is tabulated in table 2. The total amounts (%) of drug released per weight of (p)-MCM-41 powder ($R_{CDM}$) and amounts (%) of CDM remaining in (p)-MCM-41-CDM calculated using the equations (2) and (3) are shown in table 1. The loading content and adsorption efficiency of CDM in the materials were found to be dependent on the surface chemistry of the nanoparticles. Seemingly, a substantially greater quantity of CDM could be integrated into p-MCM-41, compared with that of MCM-41, suggesting that the enhanced surface polarity of p-MCM-41 certainly facilitated the adsorption of CDM into p-MCM-41. Fascinatingly, the $R_{CDM}$ values of (p)-MCM-41 seemed to be marginally different. Hence, the plasma treatment of MCM-41 not only explicitly boosted the CDM loading capability, but also helped retain CDM in the nanoparticles.

The concentrations of CDM released versus time of (p)-MCM-41-CDM-CMCS analysed by HPLC are also tabulated in table 2. All three (p)-MCM-41-CDM-CMCS hydrogels provided prolonged drug releases up to 10 days with rapid burst releases of CDM observed on day 1 of measurement. The initial burst drug release was typically seen in most drug carriers owing to an osmotic effect [44]. In these particular, CDM-loaded composite hydrogels, it was likely caused by the quick diffusion of

CDM highly absorbed at the outer surfaces of (p)-MCM-41 and also in the CMCS matrices resulting from the leaching of loosely bound CDM in the mesoporous material upon the mixing of CDM-loaded nanoparticles and CMCS solution. Not only the similar release profiles, but also the almost comparable CDM concentrations detected daily were surprisingly noted in the (p)-MCM-41-CDM-CMCS hydrogels prepared from the same amount of CDM initially exploited. Thus, the CDM content remaining in p-MCM-41-200CDM-CMCS was supposedly larger than that in MCM-41-200CDM-CMCS. The total amounts of CDM released from both MCM-41-200CDM-CMCS and p-MCM-41-200CDM-CMCS were quite close.

In addition to the appropriate physical and chemical structure of (p)-MCM-41 that could assist the controlled release of CDM from the composite hydrogels, the interaction between CDM and CMCS molecules, if it ever happened, could also help sustain the drug release. The results of the *in vitro* stability of CDM-loaded composite hydrogels demonstrated that the incorporation of CDM-loaded nanoparticles into CMCS significantly slowed down the degradation of the hydrogel matrix. Moreover, the stability of the composite hydrogels slightly increased with an increasing CDM concentration initially loaded. FT-IR-ATR analysis was employed to analyse an interaction between the CDM and CMCS molecules after the steam treatment. The FT-IR spectrum of 200CDM-CMCS sample exclusively indicated an intramolecular crosslinking between amine and carboxylate groups in CMCS and an intermolecular crosslinking between amine groups in CDM and carboxylate groups in CMCS. The amine groups in CDM were probably generated from the chain scission reaction at the amide groups of some CDM upon the steam treatment at 105°C for 10 min. The HPLC chromatograms shown in electronic supplementary material, figure S5 also supported the partial degradation of CDM at elevated temperatures. This was consistent with a previous study by Oesterling *et al.* [45] who reported the thermal degradation of CDM via amide hydrolysis. In (p)-MCM-41-CDM-CMCS, since the molecules of CDM physically adsorbed in (p)-MCM-41 could form H-bonding with the nanoparticles; upon crosslinking with steam, the molecules of all substances were, therefore, intermolecularly linked together, enhancing the stability of the composite hydrogels.

CDM released from composite hydrogels retained its antibacterial function against *Streptococcus sanguinis*, which is one of the most common microorganisms found in jaw bone infection [1]. Its good bone penetration will further facilitate the distribution of the released CDM into poor vascularized areas of the infected jaw bone [4,5]. CDM possesses bacteriostatic activity and at high-dose bactericidal activity over a wide range of pathogenic bacteria [46]. Both effects of CDM have proven to be of clinical significance. Some infections with a high bacterial load may resist bactericidal antibacterial agents, including the widely used drug penicillin [47,48]. However, the bacteriostatic activity of CDM is proven to be clinically effective in these conditions since this bacteriostatic agent, but not the bactericidal penicillin, inhibits protein synthesis in resting slow-growing bacteria found in high bacterial load infections. The CDM-loaded composite hydrogels developed in the present study possessed initially high bactericidal activity followed by continuing bacteriostatic activity for up to at least 14 days, a minimum time course required to treat jaw bone infections [1,4].

The levels of CDM released from the composite hydrogels after day 10 might be too low to be detected by HPLC (less than 5–6 µg ml$^{-1}$). Such concentrations are still higher than the minimum inhibitory concentration (MIC) and minimum bactericidal concentration (MBC) of CDM against *Streptococcus sanguinis*, which were 0.05 and 0.1 µg ml$^{-1}$, respectively (data not shown). CDM at subinhibitory concentrations, i.e. below 0.05 µg ml$^{-1}$, was also effective against some bacteria species by reducing bacterial adherence to bone surfaces [49]. In addition to delivery of CDM, the SEM results (figure 5c) indicated that the surfaces of (p)-MCM-41-200CDM-CMCS hydrogels possessed strong antibacterial activity, preventing bacterial adhesion and possible biofilm formation on the hydrogel surfaces. Bacterial cells in mature biofilm are more resistant to antibiotics than planktonic bacteria (non-adherent cells) and often lead to persistent infections [50]. The interaction of CDM-loaded MCM-41 nanoparticles to the bacterial cell wall may enable high local CDM concentration [51], which is more efficient in killing bacteria and, thus, sustaining the antibacterial efficiency of the composite hydrogels. The antibacterial surfaces might also be attributed to their negatively charged surfaces that potentially prevent the adhesion to the negatively charged bacterial cell wall [52]. Further studies are necessary to gain more in-depth knowledge to improve their clinical benefits.

By contrast, a marked reduction in the antibacterial efficiency of p-MCM-41-50CDM-CMCS on day 8 may support an important role of CDM-loaded nanoparticle-mediated antibacterial surface since its released CDM levels appeared comparable to those detected in p-MCM-41-200CDM-CMCS. It is

reasonable to expect that the level of functionally active CDM available and exposed to bacteria on the hydrogel surface was much lower in p-MCM-41-50CDM-CMCS, compared with the other.

Major disadvantages of oral CDM include the development of antibiotic-associated diarrhea, antibiotic-resistant bacteria, and dysbiosis [7,9–11]. An appropriate use of local delivery of antibiotics in jaw bone surgery procedures, therefore, helps lower these disadvantages. This emphasizes the importance of effective local delivery of antibacterial agents such as p-MCM-41-200CDM-CMCS developed in this study. It is noteworthy that high doses of oral administration of CDM, i.e. 1200–1800 mg day$^{-1}$, for 7–14 days are normally needed for treatment of jaw bone infection [8]. This results in a total CDM of 16.8–25.2 g given to individuals. Similar antibacterial activity is expected in p-MCM-41-200CDM-CMCS which is 84–126 times less drug used with an added value of osteogenic stimulation. This promising result requires further *in vivo* studies to confirm its clinical benefit. Moreover, a longer duration of antibacterial activity may be required in chronic bone infection, such as chronic osteomyelitis. Additional modifications of the composite hydrogel, such as chemical surface functionalization of MCM-41 with different organic groups [53], are potential strategies to reduce an initial drug burst and sustain the slow release of the antibiotic.

The increased ALP activity induced by p-MCM-41-200CDM-CMCS prompted us to investigate the mechanism that could potentially control this effect. The stimulation of ALP activity by the tested hydrogel appeared to be mediated by p38 MAPK. MAPKs play crucial roles as mediators of cellular responses to various extracellular stimuli [54] and have also been used to regulate ALP activity [55–58]. Among three major subfamilies of MAPKs, p38 MAPK was found to be the most critical pathway for control of ALP expression and activity in differentiating preosteoblasts [38–40]. It is likely that p-MCM-41-200CDM-CMCS enhanced mineralization by, at least partly, increasing the ALP activity, but not by stimulating the proliferation and osteoblast differentiation of hMSCs. It remains unclear which components are responsible for this. An unexpected role of CDM in stimulating bone regeneration *in vivo* and the stimulatory effect of chitosan derivatives on ALP activity and mineralization have previously been suggested [59–61]. Upon cellular uptake, silica nanoparticles may mediate intracellular signallings that activate the expression of ALP gene and/or the activity of ALP [62,63]. The released CDM and degradation products of the tested hydrogel might play some roles, and further studies are needed to confirm this.

# 5. Conclusions

CDM-loaded composite hydrogels composed of MCM-41 nanoparticles and biodegradable CMCS were successfully developed for treatment of jaw bone infection. Plasma treatment of the nanoparticles prior to CDM loading was proven to enhance the drug loading capacity and sustain the drug release with a subsided burst release effect. Both composite hydrogels integrated with CDM-loaded unmodified- and surface-modified MCM-41 nanoparticles demonstrated prolonged drug release profiles up to 10 days. Furthermore, the interaction between CDM and CMCS molecules upon crosslinking by a steam treatment not only enhanced the *in vitro* stability of the composite hydrogels but also helped retain CDM on the hydrogel matrix. CDM released from the composite hydrogels, particularly p-MCM-41-200CDM-CMCS, exhibited good *in vitro* antibacterial activity against *Streptococcus sanguinis* for at least 14 days, no cytotoxicity, and osteogenic stimulation by increasing biomineralization via p38 MAPK-dependent stimulation of ALP activity. This suggested a promising implication of these antibiotic-releasing composite hydrogels in the treatment of jaw bone infection.

Data accessibility. Data are available from the Dryad Digital Repository at https://doi.org/10.5061/dryad.k0p2ngf88 [64].

Authors' contribution. P.S. contributed to design, acquisition, analysis and interpretation of data and drafted the article. B.T. contributed to conception and design, acquisition, analysis and interpretation of data, and critically revised the article. P.K., V.P. and S.P. contributed to acquisition, analysis and interpretation of data. W.S. and W.J. contributed to conception and design, analysis and interpretation of data, and critically revised the article. All authors approved the final version of the article and agreed to be accountable for all aspects of the work in ensuring that questions related to the accuracy or integrity of any part of the work are appropriately investigated and resolved.

Competing interests. We declare we have no competing interests. The authors declared no potential conflicts of interest with respect to the research, authorship and/or publication of this article.

Funding. The present study was financially supported by the National Metal and Materials Technology Center (P1650068 (MT-B-59-BMD-13-222-G)) and Thailand Science Research and Innovation Fundamental Fund (Project no. 2323444), Thailand.

Acknowledgements. The authors are grateful to Ms. Siriporn Kitchaicharoenporn from National Metal and Materials Technology Center (MTEC) and Ms. Wanwisa Srinuanchai from National Nanotechnology Center (NANOTEC) for their assistance with TGA-DTA and HPLC experiments, respectively. The study was supported by Thammasat University Research Unit in Mineralized Tissue Reconstruction.

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
