## [Peer Review File · Royal Society Open Science]

Review History

RSOS-210808.R0 (Original submission)

Review form: Reviewer 1

Is the manuscript scientifically sound in its present form?

Yes

Are the interpretations and conclusions justified by the results?

Yes

Is the language acceptable?

Yes

Do you have any ethical concerns with this paper?

No

Have you any concerns about statistical analyses in this paper?

No

Recommendation?

Accept with minor revision (please list in comments)

Comments to the Author(s)

This manuscript reports on a mesoporous silica nanoparticle/carboxymethyl chitosan composite hydrogel system for clindamycin delivery for potential antibacterial and osteogenic applications. The materials are characterized by various techniques and authors' conclusions are supported by the experimental data. The referee supports its acceptance with minor revisions noted.

1. In Figure 1b, the molecular structure of CMCS is wrong - it's a polymer.
2. The MCM-41 was illustrated as hexagonal structure, but presented as core-shell spherical structure when embedded in CMCS matrix. It can be confusing - please revise and make it consistent.
3. Please provide the size and distribution profile of the MCM-41 prepared in this study in the Supporting Information.
4. The incorporation of MCM-41-CDM greatly enhanced the stability of CMCS hydrogel (Figure 3). It is quite interesting, and the authors' explanation is reasonable. To further test authors' hypothesis, I would suggest preparing free CDM-loaded CMCS hydrogel and testing its stability.

Review form: Reviewer 2**Is the manuscript scientifically sound in its present form?**

Yes

Are the interpretations and conclusions justified by the results?

Yes

Is the language acceptable?

Yes

Do you have any ethical concerns with this paper?

No

Have you any concerns about statistical analyses in this paper?

No

Recommendation?

Accept with minor revision (please list in comments)

Comments to the Author(s)

1. Figure S3(a) Y-axis is weight lose. I think it can change to remain weight
2. in Page 19 form the table 2 data the p-MCM-41-50CDM-CMCS release less CDM than the p-MCM-41-200CDM-CMCS but there is no data of the total amounts of drug actually loaded, released, and remaining per weight of CDM-loaded p-MCM-41-50CDM in table 1, so it is no sufficient to compare.
3. On page 20 to explore the effect of CDM on the stability of p-MCM-41-CDM-CMCS hydrogels, why use specimen of CDM-loaded CMCS hydrogel was subjected to a 3-day drug release procedure? Is there significant difference between the original hydrogel and 3day drug released one on FTIR spectra?

Decision letter (RSOS-210808.R0)

Dear Dr Singhatanadgit:

Title: Antibacterial and Osteogenic Activities of Clindamycin-releasing Mesoporous Silica/Carboxymethyl Chitosan Composite Hydrogels
Manuscript ID: RSOS-210808

Thank you for submitting the above manuscript to Royal Society Open Science. On behalf of the Editors and the Royal Society of Chemistry, I am pleased to inform you that your manuscript will be accepted for publication in Royal Society Open Science subject to minor revision in accordance with the referee suggestions. Please find the reviewers' comments at the end of this email.

The reviewers and handling editors have recommended publication, but also suggest some minor revisions to your manuscript. Therefore, I invite you to respond to the comments and revise your manuscript.

Because the schedule for publication is very tight, it is a condition of publication that you submit the revised version of your manuscript before 28-Jul-2021. Please note that the revision deadline will expire at 00.00am on this date. If you do not think you will be able to meet this date please let me know immediately.

- 1) A text file of the manuscript (tex, txt, rtf, docx or doc), references, tables (including captions) and figure captions. Do not upload a PDF as your "Main Document".
- 2) A separate electronic file of each figure (EPS or print-quality PDF preferred (either format should be produced directly from original creation package), or original software format)
- 3) Included a 100 word media summary of your paper when requested at submission. Please ensure you have entered correct contact details (email, institution and telephone) in your user account
- 4) Included the raw data to support the claims made in your paper. You can either include your data as electronic supplementary material or upload to a repository and include the relevant doi within your manuscript
- 5) All supplementary materials accompanying an accepted article will be treated as in their final form. Note that the Royal Society will neither edit nor typeset supplementary material and it will

be hosted as provided. Please ensure that the supplementary material includes the paper details where possible (authors, article title, journal name).

Kind regards,
Dr Laura Smith
Publishing Editor, Journals

On behalf of the Subject Editor Professor Anthony Stace and the Associate Editor Professor Chaohua Cui.

RSC Associate Editor:
Comments to the Author:
(There are no comments.)

RSC Subject Editor:
Comments to the Author:
(There are no comments.)

Reviewer comments to Author:
Reviewer: 1
Comments to the Author(s)

This manuscript reports on a mesoporous silica nanoparticle/carboxymethyl chitosan composite hydrogel system for clindamycin delivery for potential antibacterial and osteogenic applications. The materials are characterized by various techniques and authors' conclusions are supported by the experimental data. The referee supports its acceptance with minor revisions noted.

1. In Figure 1b, the molecular structure of CMCS is wrong - it's a polymer.
2. The MCM-41 was illustrated as hexagonal structure, but presented as core-shell spherical structure when embedded in CMCS matrix. It can be confusing - please revise and make it consistent.
3. Please provide the size and distribution profile of the MCM-41 prepared in this study in the Supporting Information.

4. The incorporation of MCM-41-CDM greatly enhanced the stability of CMCS hydrogel (Figure 3). It is quite interesting, and the authors' explanation is reasonable. To further test authors' hypothesis, I would suggest preparing free CDM-loaded CMCS hydrogel and testing its stability.

Reviewer: 2

Comments to the Author(s)

1. Figure S3(a) Y-axis is weight lose. I think it can change to remain weight

2. in Page 19 from the table 2 data the p-MCM-41-50CDM-CMCS release less CDM than the p-MCM-41-200CDM-CMCS but there is no data of the total amounts of drug actually loaded, released, and remaining per weight of CDM-loaded p-MCM-41-50CDM in table 1, so it is not sufficient to compare.

3. On page 20 to explore the effect of CDM on the stability of p-MCM-41-CDM-CMCS hydrogels, why use specimen of CDM-loaded CMCS hydrogel was subjected to a 3-day drug release procedure? Is there significant difference between the original hydrogel and 3day drug released one on FTIR spectra?

Author's Response to Decision Letter for (RSOS-210808.R0)

See Appendix A.

Decision letter (RSOS-210808.R1)

Dear Dr Singhatanadgit:

Title: Antibacterial and Osteogenic Activities of Clindamycin-releasing Mesoporous Silica/Carboxymethyl Chitosan Composite Hydrogels
Manuscript ID: RSOS-210808.R1

It is a pleasure to accept your manuscript in its current form for publication in Royal Society Open Science. The chemistry content of Royal Society Open Science is published in collaboration with the Royal Society of Chemistry.

===COVID-SPECIFIC TEXT -- WILL ONLY BE ADDED TO COVID-PAPERS BY THE EDITORIAL OFFICE===

COVID-19 rapid publication process:

We are taking steps to expedite the publication of research relevant to the pandemic. If you wish, you can opt to have your paper published as soon as it is ready, rather than waiting for it to be published the scheduled Wednesday.

This means your paper will not be included in the weekly media round-up which the Society sends to journalists ahead of publication. However, it will still appear in the COVID-19

Publishing Collection which journalists will be directed to each week
(<https://royalsocietypublishing.org/topic/special-collections/novel-coronavirus-outbreak>).

If you wish to have your paper considered for immediate publication, or to discuss further, please notify openscience_proofs@royalsociety.org and press@royalsociety.org when you respond to this email.

===END OF COVID-SPECIFIC TEXT -- WILL BE REMOVED AS NECESSARY BY THE EDITORIAL OFFICE===

Yours sincerely,
Dr Ellis Wilde
Publishing Editor, Journals

On behalf of the Subject Editor Professor Anthony Stace and the Associate Editor Professor Chaohua Cui.

RSC Associate Editor
Comments to the Author:
(There are no comments.)

Reviewer(s)' Comments to Author:

Appendix A

Responses to the Reviewers' Comments

Journal: Royal Society Open Science

Manuscript #: RSOS-210808

Title of Paper: Antibacterial and Osteogenic Activities of Clindamycin-releasing Mesoporous Silica/Carboxymethyl Chitosan Composite Hydrogels

Authors: Piyarat Sungkhaphan, Boonlom Thavornyutikarn, Pakkanun Kaewkong, Veerachai Pongkittiphan, Soraya Pornsuwan, Weerachai Singhatanadgit, Wanida Janvikul

Date Sent: July 23rd, 2021

We are thankful to the Editor and Reviewers for your time and efforts to consider and review our revised manuscript. We have edited our manuscript according to the Reviewers' comments. We believe that the revised version now meets the journal publication requirements. Below are our responses to the Reviewers' comments.

Reviewer 1

Comments to the Author(s)

This manuscript reports on a mesoporous silica nanoparticle/carboxymethyl chitosan composite hydrogel system for clindamycin delivery for potential antibacterial and osteogenic applications. The materials are characterized by various techniques and authors' conclusions are supported by the experimental data. The referee supports its acceptance with minor revisions noted.

Comment 1: In Figure 1b, the molecular structure of CMCS is wrong - it's a polymer.

Response: The molecular structure of CMCS depicted in Figure 1b has been corrected, as suggested.

Comment 2: The MCM-41 was illustrated as hexagonal structure, but presented as core-shell spherical structure when embedded in CMCS matrix. It can be confusing - please revise and make it consistent.

Response: As suggested, the MCM-41 embedded in CMCS matrix, depicted in Figure 1b, has been revised in the hexagonal structure form, as shown in the revised manuscript and below.

Comment 3: Please provide the size and distribution profile of the MCM-41 prepared in this study in the Supporting Information.

Response: We have not yet determined the size and distribution profile of the MCM-41 material, primarily due to its high specific surface area and tendency to form agglomerates which ultimately leads to a difficulty in the evaluation of the actual size and distribution of the material. Serving as a drug carrier, the MCM-41 material was, instead, characterized in terms of pore size and distribution profile by BET analysis as this property was relevantly related to the drug loading and release capacity. The BET results are originally reported in Section 3.1 on page 16 in the revised manuscript and plotted in Figure S1 in Supporting Information. Anyhow, to accomplish the complete physical information of the material, the particle size and morphology of the MCM-41 material should be investigated, as suggested; this can be conducted soon after the COVID-19 crisis becomes relieved in Thailand, especially in Bangkok and vicinity.

Comment 4: The incorporation of MCM-41-CDM greatly enhanced the stability of CMCS hydrogel (Figure 3). It is quite interesting, and the authors' explanation is reasonable. To further test authors' hypothesis, I would suggest preparing free CDM-loaded CMCS hydrogel and testing its stability.

Response: Thank you very for the suggestion. Certainly, we have already tested the stability of the free CDM-loaded CMCS hydrogel. It was found that the free CDM-loaded CMCS hydrogel exhibited good stability for more than 30 days of incubation time.

Reviewer 2

Comments to the Author(s)

Comment 1: Figure S3(a) Y-axis is weight loss. I think it can change to remain weight

Response: As suggested, the Y-axis of TGA thermograms in Figure S3(a) has been changed to % remaining weight.

Comment 2: In Page 19 form the table 2 data the p-MCM-41-50CDM-CMCS release less CDM

than the p-MCM-41-200CDM-CMCS but there is no data of the total amounts of drug actually loaded, released, and remaining per weight of CDM-loaded p-MCM-41-50CDM in table 1, so it is no sufficient to compare.

Response: Thank you for the comment. In the revised manuscript, we have added the information about the total amounts of drug actually loaded, released, and remaining per weight of CDM-loaded-p-MCM-41-50CDM in the revised Table 1 (Page 42).

Comment 3: On page 20 to explore the effect of CDM on the stability of p-MCM-41-CDM-CMCS hydrogels, why use specimen of CDM-loaded CMCS hydrogel was subjected to a 3-day drug release procedure? Is there significant difference between the original hydrogel and 3 day drug released one on FTIR spectra?

Response: We truly appreciate the Reviewer's comment. We were, of course, aware of this issue, too, and comparatively conducted the FTIR analysis of the original hydrogel. As revealed in Figure S4, a newly added Figure in the revised SI, there was a difference in the FT-IR spectra of :

Section 3 (Page 11) in the revised manuscript.

Figure S4. Overlaid FT-IR spectra of CDM-loaded CMCS hydrogel: (a) before and (b) after 3 days of CDM release.